# The Monetary Value of Human Lives Lost to Suicide in the African Continent: Beating the African War Drums

**DOI:** 10.3390/healthcare8020084

**Published:** 2020-04-02

**Authors:** Joses M. Kirigia, Rosenabi D.K. Muthuri, Newton G. Muthuri

**Affiliations:** 1Department of Research, African Sustainable Development Research Consortium (ASDRC), Nairobi 00100, Kenya; 2Faculty of Health Sciences, University of Pretoria, Hatfield, Pretoria 0002, South Africa; u19391189@tuks.co.za; 3School of Business, United States International University, P. O. Box 14634-00800, Nairobi 00800, Kenya; nmuthuri@usiu.ac.ke

**Keywords:** suicide deaths, human capital approach, monetary value of human lives, non-health gross domestic product

## Abstract

*Background:* Suicide is an important public health problem in the African continent whose economic burden remains largely unknown. This study estimated the monetary value of human lives lost due to suicide in the African continent in 2017. *Methods:* The human capital approach was applied to monetarily value the years of life lost due to premature mortality from suicide deaths (SD) among 54 African countries. A 3% discount rate was used to convert future losses into their present values. The sensitivity of monetary value of human lives lost to changes in discount rate and average life expectancy was tested. *Results:* The 75,505 human lives lost from suicide had a grand total monetary value of International Dollars (Int$) 6,989,963,325; and an average present value of Int$ 92,576 per SD. About 31.1% of the total monetary value of SD was borne by high-income and upper-middle-income countries (Group 1); 54.4% by lower-middle-income countries (Group 2); and 14.5% by low-income countries (Group 3). The average monetary value per human life lost from SD was Int$ 234,244 for Group 1, Int$ 109,545 for Group 2 and Int$ 32,223 for Group 3. *Conclusions:* Evidence shows that suicide imposes a substantive economic burden on African economies. The evidence reinforces the case for increased investments to ensure universal coverage of promotive, preventive, curative and rehabilitative mental health services.

## 1. Introduction

The African continent has 54 countries: 1 (1.9%) high-income, 8 (14.8%) upper-middle-income, 21 (38.9%) lower-middle-income and 24 (44.4%) low-income countries (see Table 1) [1]. The continent had a population of around 1,277,972,481 people and a gross domestic product (GDP) of International Dollars (Int$) 6,062,729,000,000 in 2019 [2].

The United States Centers for Disease Prevention and Control (CDC) defines suicide as “death caused by self-directed injurious behaviour with an intent to die as a result of the behaviour (p. 23)” [3]. Globally, an estimated total of 793,823 deaths from self-harm occurred in 2017, of whom 540,226 (68.1%) were males and 253,597 (31.9%) females [4]. Thus, men die by suicide 2.1 times more often than women. Approximately 75,505 (9.51%) of those deaths were borne by the African continent.

The number of suicide deaths varied from a minimum of 5 in São Tomé and Príncipe to a maximum of 8069 in Nigeria. Thirty-two (59.3%) countries had less than 1000 suicide deaths, 10 (18.5%) countries had 1000–2000 deaths, 6 (11.1%) countries had 2001–3000 deaths, and 6 (11.1%) countries had over 3000 suicide deaths. Figure 1 below shows the distribution of suicide deaths across the 54 countries in the continent.

Despite the limited epidemiological data, available global burden of disease estimates indicates suicidal behaviour is an important public health problem in Africa [3]. Various factors increase a person’s risk for attempting or dying by suicide, e.g., family history of suicide, violent relationship, paucity of supportive relationships, barriers to health care (e.g., lack of access to providers or medications and stigma), availability of lethal means of suicide, unsafe media portrayals of suicide [5], previous suicide attempt(s), loss of self-esteem, hopelessness, stress, loneliness, physical illness, substance abuse (including alcohol), physical sickness, history of mental illness such as anxiety and depression [6], economic recession with loss of source of livelihood [7,8,9,10,11,12,13] and unemployment [14,15].

The United Nations (UN) General Assembly Resolution A/RES/70/1, which was adopted in September 2015, Sustainable Development Goal (SDG) 3 focusses on ensuring healthy lives and promoting well-being for all people (including those prone to suicide) at all ages [16]. In line with this goal, the World Health Organization (WHO) and the World Bank developed a universal health coverage (UHC) index to monitor the average coverage of essential health services based on tracer interventions that include reproductive, maternal, new born and child health, infectious diseases, noncommunicable diseases (including mental and neurological disorders) and service capacity and access [17].

The UHC service coverage gap was less than 30% in 1 country; 30–40% in 9 countries; 41–50% in 8 countries; 51–60% in 17 countries; 61–70% in 17 countries; and above 70% in two countries. Figure 2 shows the UHC indices and gaps for individual African countries in 2015 [18].

The sub-optimal essential health services coverage could be partially attributed to under-investment in national health systems among various countries. Only Cabo Verde, Mauritius, Morocco, Seychelles, Sudan, Swaziland and Tunisia had per capita health spending greater than the level of expenditure recommended by Stenberg et al. [19] for achieving SDG3. Out of the remaining 47 (87%) countries, 2 (4%) had a health spending deficit of United States Dollar (US$) 10–30; 5 (11%) had a deficit of US$ 31–60; 25(53%) a deficit of US$ 61–90; 6 (13%) had a deficit of US$ 90 –100; and 9 countries had a deficit of over US$ 100. Figure 3 shows the current health expenditure (CHE) per capita and deficit in health spending per capita in Africa in 2016 [20].

Suicide is an important public health problem in the African continent whose actual epidemiological and economic burden remains largely unknown due to weak surveillance systems [21]. Various studies have attempted to estimate the cost of suicide in the Americas [22,23,24,25,26,27,28], Europe [29,30,31,32,33,34,35,36,37,38,39], Western Pacific [40,41,42,43,44,45,46,47,48,49,50,51], and Southeast Asia [52] for use in raising the public and policy-maker’s awareness. 

To date, no study has attempted to estimate the monetary value of human lives lost due to suicide in the African continent. The study reported in this paper attempted to contribute to bridging that knowledge gap. Such economic evidence would be useful for advocacy for prioritisation of suicide prevention on the continental, sub-regional and national development and public health policy agendas, and to raise public awareness of suicide as an important public health challenge.

This paper addresses the following research question: What is the monetary value of human lives lost due to suicide in the African continent? The current study estimated the monetary value of human lives lost due to suicide in the African continent in 2017.

## 2. Materials and Methods 

### 2.1. Study Area and Population

Our study included all 54 countries of Africa. It focussed on the monetary valuation of the 75,505 human lives lost from suicide (intentional self-harm) in 2017.

### 2.2. Analytical Framework

The human capital approach (HCA) was used to estimate economic value of years of life lost (YLL) to premature mortality from suicide [53]. The approach has been in use since ground-breaking research of Petty in the 16th century [54] and Farr [55] in the 18th century. A historical evolution of HCA was provided by Kiker [56]. Fein [57], Mushkin and Collings [58], Weisbrod [59], and Landefeld and Seskin [60] developed theoretical foundations of the HCA. Linnerooth [61] and Mooney [62] highlighted strong points and flaws of the HCA. The form of HCA employed to monetarily value years of life lost is akin to that used in estimating the productivity losses associated with diabetes mellitus [63], cholera [64], maternal deaths in 2010 [65] and 2013 [66], child mortality [67], Ebola virus disease [68,69], tuberculosis [70], non-communicable diseases [71] and neglected tropical diseases [72] in Africa.

Consistent with the abovementioned past studies, GDP per person was used to value YLL into money [73]. GDP is made up of investment expenditure, consumption expenditure, government expenditure, and net exports (exports minus imports) [74]. Premature death of an individual from any cause depletes GDP since the dead do not save, invest, consume, pay taxes, or produce commodities.

We followed Weisbrod’s [75] counsel below in quantifying expected money value to society of human lives lost due to suicide: 


*“Such a definition (of society) leads to the proposition that the value of a person to others is measured by any excess of his contribution to production over what he consumes from production—this difference being the amount by which everyone else benefits from his productivity. … The present value of a man at any given age may be defined operationally as his discounted expected future earnings stream (net of his consumption if the net concept is used)”*
(pp. 426–427)

Weisbrod [75], World Health Organization (WHO) [76] and Chisholm et al. [77] agreed that the quantity of interest should be the effect of premature mortality on the total national expenditure net of spending on health.

The total financial value of YLL (TMVYLL) from suicide in Africa is equal to the sum of potential non-health GDP losses of the 54 countries [63,64,65,66,67,68,69,70,71,72]. The TMVYLL of deaths from suicide in a country is the sum of the potential non-health GDP lost due to suicide among people aged 10–14 years $\left(MVYLL_{10-14}\right)$, 15–19 years $\left(MVYLL_{15-19}\right)$, 20–24 years $\left(MVYLL_{20-24}\right)$, 25–29 years $\left(MVYLL_{25-29}\right)$, 30–34 years $\left(MVYLL_{30-34}\right)$, 35–39 years $\left(MVYLL_{35-39}\right)$, 40–44 years $\left(MVYLL_{40-44}\right)$, 45–49 years $\left(MVYLL_{45-49}\right)$, 50–54 years $\left(MVYLL_{50-54}\right)$, 55–59 years $\left(MVYLL_{55-59}\right)$, 60–64 years $\left(MVYLL_{60-64}\right)$, 65–69 years $\left(MVYLL_{65-69}\right)$, 70–74 years $\left(MVYLL_{70-74}\right)$, 75–79 years $\left(MVYLL_{75-79}\right)$, 80–84 years $\left(MVYLL_{80-84}\right)$, 85–89 years $\left(MVYLL_{85-89}\right)$, 90–94 years $\left(MVYLL_{90-94}\right)$, and 95 years and above $\left(MVYLL_{\geq 95}\right)$.

The $MVYLL_{k}$ related with suicide deaths (SDs) among persons of k^th^ age group was obtained by multiplying total discounted YLL, per-capita non-health GDP in international dollars (or purchasing power parity) and the total number of SDs in the age group [63,64,65,66,67,68,69,70,71,72]. Each of the 54 country’s discounted monetary value of human lives loss linked with SDs was estimated using Equations presented below:(1)$$TMVYLL=\left(\begin{array}{l} MVYLL_{10-14}+MVYLL_{15-19}+MVYLL_{20-24}+\ldots +\\ MVYLL_{80-84}+MVYLL_{85-89}+MVYLL_{90-94}+MVYLL_{\geq 95} \end{array}\right)$$
(2)$$\begin{array}{l} MVYLL_{10-14}=\sum_{i=1}^{k}\{\left[1/{\left(1+r\right)}^{k}\right]\times \left[NGDPC_{Int\$}\right]\times \left[SD_{10-14}\right]\}=\\ \;\;\{\left[1/{\left(1+r\right)}^{1}\right]\times \left[NGDPC_{Int\$}\right]\times \left[SD_{10-14}\right]\}+\\ \;\;\{\left[1/{\left(1+r\right)}^{2}\right]\times \left[NGDPC_{Int\$}\right]\times \left[SD_{10-14}\right]\}+\ldots +\\ \;\;\{\left[1/{\left(1+r\right)}^{k}\right]\times \left[NGDPC_{Int\$}\right]\times \left[SD_{10-14}\right]\} \end{array}$$
(3)$$\begin{array}{l} MVYLL_{15-19}=\sum_{i=1}^{k}\{\left[1/{\left(1+r\right)}^{k}\right]\times \left[NGDPC_{Int\$}\right]\times \left[SD_{15-19}\right]\}=\\ \;\;\{\left[1/{\left(1+r\right)}^{1}\right]\times \left[NGDPC_{Int\$}\right]\times \left[SD_{15-19}\right]\}+\\ \;\;\{\left[1/{\left(1+r\right)}^{2}\right]\times \left[NGDPC_{Int\$}\right]\times \left[SD_{15-19}\right]\}+\ldots +\\ \;\;\{\left[1/{\left(1+r\right)}^{k}\right]\times \left[NGDPC_{Int\$}\right]\times \left[SD_{15-19}\right]\} \end{array}$$
(4)$$\begin{array}{l} MVYLL_{20-24}=\sum_{i=1}^{k}\{\left[1/{\left(1+r\right)}^{k}\right]\times \left[NGDPC_{Int\$}\right]\times \left[SD_{20-24}\right]\}=\\ \;\;\{\left[1/{\left(1+r\right)}^{1}\right]\times \left[NGDPC_{Int\$}\right]\times \left[SD_{20-24}\right]\}+\\ \;\;\{\left[1/{\left(1+r\right)}^{2}\right]\times \left[NGDPC_{Int\$}\right]\times \left[SD_{20-24}\right]\}+\ldots +\\ \;\;\{\left[1/{\left(1+r\right)}^{k}\right]\times \left[NGDPC_{Int\$}\right]\times \left[SD_{20-24}\right]\} \end{array}$$
(5)$$\begin{array}{l} MVYLL_{25-29}=\sum_{i=1}^{k}\{\left[1/{\left(1+r\right)}^{k}\right]\times \left[NGDPC_{Int\$}\right]\times \left[SD_{25-29}\right]\}=\\ \;\;\{\left[1/{\left(1+r\right)}^{1}\right]\times \left[NGDPC_{Int\$}\right]\times \left[SD_{25-29}\right]\}+\\ \;\;\{\left[1/{\left(1+r\right)}^{2}\right]\times \left[NGDPC_{Int\$}\right]\times \left[SD_{25-29}\right]\}+\ldots +\\ \;\;\{\left[1/{\left(1+r\right)}^{k}\right]\times \left[NGDPC_{Int\$}\right]\times \left[SD_{25-29}\right]\} \end{array}$$
(6)$$\begin{array}{l} MVYLL_{30-34}=\sum_{i=1}^{k}\{\left[1/{\left(1+r\right)}^{k}\right]\times \left[NGDPC_{Int\$}\right]\times \left[SD_{30-34}\right]\}=\\ \;\;\{\left[1/{\left(1+r\right)}^{1}\right]\times \left[NGDPC_{Int\$}\right]\times \left[SD_{30-34}\right]\}+\\ \;\;\{\left[1/{\left(1+r\right)}^{2}\right]\times \left[NGDPC_{Int\$}\right]\times \left[SD_{30-34}\right]\}+\ldots +\\ \;\;\{\left[1/{\left(1+r\right)}^{k}\right]\times \left[NGDPC_{Int\$}\right]\times \left[SD_{30-34}\right]\} \end{array}$$
(7)$$\begin{array}{l} MVYLL_{35-39}=\sum_{i=1}^{k}\{\left[1/{\left(1+r\right)}^{k}\right]\times \left[NGDPC_{Int\$}\right]\times \left[SD_{35-39}\right]\}=\\ \;\;\{\left[1/{\left(1+r\right)}^{1}\right]\times \left[NGDPC_{Int\$}\right]\times \left[SD_{35-39}\right]\}+\\ \;\;\{\left[1/{\left(1+r\right)}^{2}\right]\times \left[NGDPC_{Int\$}\right]\times \left[SD_{35-39}\right]\}+\ldots +\\ \;\;\{\left[1/{\left(1+r\right)}^{k}\right]\times \left[NGDPC_{Int\$}\right]\times \left[SD_{35-39}\right]\} \end{array}$$
(8)$$\begin{array}{l} MVYLL_{40-44}=\sum_{i=1}^{k}\{\left[1/{\left(1+r\right)}^{k}\right]\times \left[NGDPC_{Int\$}\right]\times \left[SD_{40-44}\right]\}=\\ \;\;\{\left[1/{\left(1+r\right)}^{1}\right]\times \left[NGDPC_{Int\$}\right]\times \left[SD_{40-44}\right]\}+\\ \;\;\{\left[1/{\left(1+r\right)}^{2}\right]\times \left[NGDPC_{Int\$}\right]\times \left[SD_{40-44}\right]\}+\ldots +\\ \;\;\{\left[1/{\left(1+r\right)}^{k}\right]\times \left[NGDPC_{Int\$}\right]\times \left[SD_{40-44}\right]\} \end{array}$$
(9)$$\begin{array}{l} MVYLL_{45-49}=\sum_{i=1}^{k}\{\left[1/{\left(1+r\right)}^{k}\right]\times \left[NGDPC_{Int\$}\right]\times \left[SD_{45-49}\right]\}=\\ \;\;\{\left[1/{\left(1+r\right)}^{1}\right]\times \left[NGDPC_{Int\$}\right]\times \left[SD_{45-49}\right]\}+\\ \;\;\{\left[1/{\left(1+r\right)}^{2}\right]\times \left[NGDPC_{Int\$}\right]\times \left[SD_{45-49}\right]\}+\ldots +\\ \;\;\{\left[1/{\left(1+r\right)}^{k}\right]\times \left[NGDPC_{Int\$}\right]\times \left[SD_{45-49}\right]\} \end{array}$$
(10)$$\begin{array}{l} MVYLL_{50-54}=\sum_{i=1}^{k}\{\left[1/{\left(1+r\right)}^{k}\right]\times \left[NGDPC_{Int\$}\right]\times \left[SD_{50-54}\right]\}=\\ \;\;\{\left[1/{\left(1+r\right)}^{1}\right]\times \left[NGDPC_{Int\$}\right]\times \left[SD_{50-54}\right]\}+\\ \;\;\{\left[1/{\left(1+r\right)}^{2}\right]\times \left[NGDPC_{Int\$}\right]\times \left[SD_{50-54}\right]\}+\ldots +\\ \;\;\{\left[1/{\left(1+r\right)}^{k}\right]\times \left[NGDPC_{Int\$}\right]\times \left[SD_{50-54}\right]\} \end{array}$$
(11)$$\begin{array}{l} MVYLL_{55-59}=\sum_{i=1}^{k}\{\left[1/{\left(1+r\right)}^{k}\right]\times \left[NGDPC_{Int\$}\right]\times \left[SD_{55-59}\right]\}=\\ \;\;\{\left[1/{\left(1+r\right)}^{1}\right]\times \left[NGDPC_{Int\$}\right]\times \left[SD_{55-59}\right]\}+\\ \;\;\{\left[1/{\left(1+r\right)}^{2}\right]\times \left[NGDPC_{Int\$}\right]\times \left[SD_{55-59}\right]\}+\ldots +\\ \;\;\{\left[1/{\left(1+r\right)}^{k}\right]\times \left[NGDPC_{Int\$}\right]\times \left[SD_{55-59}\right]\} \end{array}$$
(12)$$\begin{array}{l} MVYLL_{60-64}=\sum_{i=1}^{k}\{\left[1/{\left(1+r\right)}^{k}\right]\times \left[NGDPC_{Int\$}\right]\times \left[SD_{60-64}\right]\}=\\ \;\;\{\left[1/{\left(1+r\right)}^{1}\right]\times \left[NGDPC_{Int\$}\right]\times \left[SD_{60-64}\right]\}+\\ \;\;\{\left[1/{\left(1+r\right)}^{2}\right]\times \left[NGDPC_{Int\$}\right]\times \left[SD_{60-64}\right]\}+\ldots +\\ \;\;\{\left[1/{\left(1+r\right)}^{k}\right]\times \left[NGDPC_{Int\$}\right]\times \left[SD_{60-64}\right]\} \end{array}$$
(13)$$\begin{array}{l} MVYLL_{65-69}=\sum_{i=1}^{k}\{\left[1/{\left(1+r\right)}^{k}\right]\times \left[NGDPC_{Int\$}\right]\times \left[SD_{65-69}\right]\}=\\ \;\;\{\left[1/{\left(1+r\right)}^{1}\right]\times \left[NGDPC_{Int\$}\right]\times \left[SD_{65-69}\right]\}+\\ \;\;\{\left[1/{\left(1+r\right)}^{2}\right]\times \left[NGDPC_{Int\$}\right]\times \left[SD_{65-69}\right]\}+\ldots +\\ \;\;\{\left[1/{\left(1+r\right)}^{k}\right]\times \left[NGDPC_{Int\$}\right]\times \left[SD_{65-69}\right]\} \end{array}$$
(14)$$\begin{array}{l} MVYLL_{70-74}=\sum_{i=1}^{k}\{\left[1/{\left(1+r\right)}^{k}\right]\times \left[NGDPC_{Int\$}\right]\times \left[SD_{70-74}\right]\}=\\ \;\;\{\left[1/{\left(1+r\right)}^{1}\right]\times \left[NGDPC_{Int\$}\right]\times \left[SD_{70-74}\right]\}+\\ \;\;\{\left[1/{\left(1+r\right)}^{2}\right]\times \left[NGDPC_{Int\$}\right]\times \left[SD_{70-74}\right]\}+\ldots +\\ \;\;\{\left[1/{\left(1+r\right)}^{k}\right]\times \left[NGDPC_{Int\$}\right]\times \left[SD_{70-74}\right]\} \end{array}$$
(15)$$\begin{array}{l} MVYLL_{75-79}=\sum_{i=1}^{k}\{\left[1/{\left(1+r\right)}^{k}\right]\times \left[NGDPC_{Int\$}\right]\times \left[SD_{75-79}\right]\}=\\ \;\;\{\left[1/{\left(1+r\right)}^{1}\right]\times \left[NGDPC_{Int\$}\right]\times \left[SD_{75-79}\right]\}+\\ \;\;\{\left[1/{\left(1+r\right)}^{2}\right]\times \left[NGDPC_{Int\$}\right]\times \left[SD_{75-79}\right]\}+\ldots +\\ \;\;\{\left[1/{\left(1+r\right)}^{k}\right]\times \left[NGDPC_{Int\$}\right]\times \left[SD_{75-79}\right]\} \end{array}$$
(16)$$\begin{array}{l} MVYLL_{80-84}=\sum_{i=1}^{k}\{\left[1/{\left(1+r\right)}^{k}\right]\times \left[NGDPC_{Int\$}\right]\times \left[SD_{80-84}\right]\}=\\ \;\;\{\left[1/{\left(1+r\right)}^{1}\right]\times \left[NGDPC_{Int\$}\right]\times \left[SD_{80-84}\right]\}+\\ \;\;\{\left[1/{\left(1+r\right)}^{2}\right]\times \left[NGDPC_{Int\$}\right]\times \left[SD_{80-84}\right]\}+\ldots +\\ \;\;\{\left[1/{\left(1+r\right)}^{k}\right]\times \left[NGDPC_{Int\$}\right]\times \left[SD_{80-84}\right]\} \end{array}$$
(17)$$\begin{array}{l} MVYLL_{85-89}=\sum_{i=1}^{k}\{\left[1/{\left(1+r\right)}^{k}\right]\times \left[NGDPC_{Int\$}\right]\times \left[SD_{85-89}\right]\}=\\ \;\;\{\left[1/{\left(1+r\right)}^{1}\right]\times \left[NGDPC_{Int\$}\right]\times \left[SD_{85-89}\right]\}+\\ \;\;\{\left[1/{\left(1+r\right)}^{2}\right]\times \left[NGDPC_{Int\$}\right]\times \left[SD_{85-89}\right]\}+\ldots +\\ \;\;\{\left[1/{\left(1+r\right)}^{k}\right]\times \left[NGDPC_{Int\$}\right]\times \left[SD_{85-89}\right]\} \end{array}$$
(18)$$\begin{array}{l} MVYLL_{90-94}=\sum_{i=1}^{k}\{\left[1/{\left(1+r\right)}^{k}\right]\times \left[NGDPC_{Int\$}\right]\times \left[SD_{90-94}\right]\}=\\ \;\;\{\left[1/{\left(1+r\right)}^{1}\right]\times \left[NGDPC_{Int\$}\right]\times \left[SD_{90-94}\right]\}+\\ \;\;\{\left[1/{\left(1+r\right)}^{2}\right]\times \left[NGDPC_{Int\$}\right]\times \left[SD_{90-94}\right]\}+\ldots +\\ \;\;\{\left[1/{\left(1+r\right)}^{k}\right]\times \left[NGDPC_{Int\$}\right]\times \left[SD_{90-94}\right]\} \end{array}$$
(19)$$\begin{array}{l} MVYLL_{\geq 95}=\sum_{i=1}^{k}\{\left[1/{\left(1+r\right)}^{k}\right]\times \left[NGDPC_{Int\$}\right]\times \left[SD_{\geq 95}\right]\}=\\ \;\;\{\left[1/{\left(1+r\right)}^{1}\right]\times \left[NGDPC_{Int\$}\right]\times \left[SD_{\geq 95}\right]\}+\\ \;\;\{\left[1/{\left(1+r\right)}^{2}\right]\times \left[NGDPC_{Int\$}\right]\times \left[SD_{\geq 95}\right]\}+\ldots +\\ \;\;\{\left[1/{\left(1+r\right)}^{k}\right]\times \left[NGDPC_{Int\$}\right]\times \left[SD_{\geq 95}\right]\} \end{array}$$
where $MVYLL_{k}$ is the monetary value of YLL among persons of k^th^ age group; $1/{\left(1+r\right)}^{k}$ is the discount factor used to converts future MVYLL into today’s dollars; r is an interest rate that measures the opportunity cost of lost output (3% in the current study); $\sum_{i=1}^{k}$ is the summation from year i to k; i is the first year of life lost; k is the final year of the total number of YLLs per SD, which is gotten by subtracting the average age at death for SD-related causes from national average life expectancy at birth; $NGDPC_{Int\$}$ is non-health GDP per capita in international dollars (or purchasing power parity), which is the difference between national GDP per capita ($GDPC_{Int\$}$) and national current health expenditure per-capita (CHEPC); $SD_{10-14}$ is the number of SDs among those aged 10–14 years in country j in 2017; $SD_{15-19}$ is the number of SDs among those aged 15–19 years in country j in 2017; $SD_{20-24}$ is the number of SDs among those aged 20–24 years in country j in 2017; $SD_{25-29}$ is the number of SDs among those aged 25–29 years in country j in 2017; $SD_{30-34}$ is the number of SDs among those aged 30–34 years in country j in 2017; $SD_{35-39}$ is the number of SDs among those aged 35–39 years in country j in 2017; $SD_{40-44}$ is the number of SDs among those aged 40–44 years in country j in 2017; $SD_{45-49}$ is the number of SDs among those aged 45–49 years in country j in 2017; $SD_{50-54}$ is the number of SDs among those aged 50–54 years in country j in 2017; $SD_{55-59}$ is the number of SDs among those aged 55–59 years in country j in 2017; $SD_{60-64}$ is the number of SDs among those aged 60–64 years in country j in 2017; $SD_{65-69}$ is the number of SDs among those aged 65–69 years in country j in 2017; $SD_{70-74}$ is the number of SDs among those aged 70–74 years in country j in 2017; $SD_{75-79}$ is the number of SDs among those aged 75–79 years in country j in 2017; $SD_{80-84}$ is the number of SDs among those aged 80–84 years in country j in 2017; $SD_{85-89}$ is the number of SDs among those aged 85–89 years in country j in 2017; $SD_{90-94}$ is the number of SDs among those aged 90–94 years in country j in 2017; and $SD_{\geq 95}$ is the number of SDs among those aged 95 years and above in country j in 2017 [63,64,65,66,67,68,69,70,71,72]. The future MVYLL were discounted to 2019 base year.

### 2.3. Data Sources

The aforementioned nineteen equations were estimated using current health expenditure per capita data for each country from the WHO Global Health Expenditure Database [20]; the average life expectancy at birth data for each country from the WHO World Health Statistics Report 2019 [18]; the GDP per capita data for each country from the International Monetary Fund (IMF) World Economic Outlook Database [2]; and the number of suicide deaths data for each country from the Institute for Health Metrics and Evaluation (IHME) Global Burden of Disease Study 2017 Database [5].

### 2.4. Data Analysis

Excel software (Microsoft, New York) was employed to perform data analysis following the twelve steps below:

Step 1: The nineteen equations presented under the analytical framework Section 2.2 were built into 54 Excel spreadsheets, i.e., one sheet per country.

Step 2: The SDs and distribution by age group per country were extracted from the IHME Global Burden of Disease Study 2017 Database (see Appendix A) [5].

Step 3: For each of the 54 countries, average life expectancy at birth data were taken from the WHO World Health Statistics Report 2019 (see Appendix A) [18].

Step 4: The number of undiscounted YLL per age group was calculated by subtracting the average age of onset of SD for the age group from each country’s average life expectancy at birth. For instance, since the life expectancy for Burkina Faso was 60.3 years and the average of age of onset of SD in age group 20-24 was 22 years, YLL = 60.3−22 = 38.3 years.

Step 5: The YLLs calculated in Step 4 were discounted at a rate of 3%.

Step 6: The GDP per capita (Int$) for each of the 54 countries was downloaded from the IMF website [2].

Step 7: The current health expenditure per person (Int$) for each of the 54 countries was downloaded from the WHO Global Health Expenditure Database [20].

Step 8: The non-health GDP per capita (Int$) for each of the 54 countries was estimated by subtracting national CHEPC from GDP per capita (see Appendix A) (File S3).

Step 9: Testing of the robustness of the results through sensitivity analysis. In the current study, we used a discount rate of 3%, which is consistent with past burden of disease [78], health systems [79] and health economics [63,64,65,66,67,68,69,70,71,72,80] studies. Two univariate sensitivity analyses were done: (1) sensitivity of MVYLL to changes in discount rate; and (2) sensitivity of MVYLL to changes in the assumed life expectancy at birth. First, to test sensitivity of MVYLL to changes in discount rate, the 19 equations were alternately recalculated with 5% and 10% discount rates. Second, the same equations were re-estimated alternately assuming average life expectancy of 76.4 years (i.e., life expectancy for Algeria) and 87 years (i.e., female life expectancy in Japan, which is the highest in the world) for all 54 African continent countries instead of their actual life expectancies to gauge the effect on the MVYLL.

Step 10: The 54 African continent countries were categorised into three economic groups as portrayed in Table 1: Group 1, 9 high/upper-middle-income countries; Group 2, 21 lower-middle-income countries; and Group 3, 24 low-income countries [1]. 

Step 11: Each of the 54 country’s population and SDs were arranged into three economic groups, i.e., Groups 1–3.

Step 12: The MVYLL results for various countries were sorted into the three groups and measures of central tendency and variability were appraised.

Step 13: The mean monetary value per SD was obtained by dividing each country’s total value of human lives lost due to SD by number of suicide deaths for that country, whereas the value of human life per person in population was calculated by dividing each country’s total value of YLL due to SD by the national population.

### 2.5. Ethics Approval

No ethical clearance was required because the study relied completely on analysis of secondary data publicly available in the IHME Website [5], IMF database [2], WHO Global Health Observatory [18], and WHO Global Health Expenditure database [20]. 

## 3. Results

### 3.1. Value of Human Life Loss Attributable to Suicide

The 75,505 human lives lost from suicide had a grand total monetary value of Int$ 6,989,963,325, which is equivalent to 0.12% of Africa’s GDP in 2019. Figure 4 portrays monetary value of the total number of SD for each of the 54 countries.

The average value of YLL lost was Int$ 92,576 per SD. The expected total value of YLL lost from suicide across the continent varied from Int$ 304,456 in Sao Tome and Principe to Int$ 1,272,197,958 in Egypt. The potential total value of YLL was under Int$ 10 million in 9 (16.7%) countries, between Int$ 10 million and Int$ 50 million in 23 (42.6%) countries, between Int$ 51 million and Int$ 100 million in 8 (14.8%) countries, and over Int$ 100 million in 14 (25.9%) countries.

Table 2 presents the potential pecuniary value of human life lost per age group.

About 11.37% of the total SD-related loss in the continent was borne by those aged 10–19 years, 29.16% by those aged 20–29 years, 37.8% by those aged 30–39 years, 13.9% by those aged 40–49 years, 6.43% by those aged 50–59 years, 1.28% by those aged 60–69 years, and 0.06% by those aged over 70 years. Thus, those in the age bracket of 15–59 years incurred 96.3% of the money value of YLL.

### 3.2. Money Value of Human Life Lost Among Group 1 Countries

The 9280 SDs in Group 1 countries led to an estimated Int$ 2,173,830,922 in value of YLL in 2019, which was equal to 0.14% of the group’s total GDP. The total value of YLL was wide-ranging, from Int$ 23,654,633 in Seychelles to Int$ 1,260,115,633 in South Africa. About 82.8% of Group 1 money value of YLL was sustained by Algeria and South Africa. Figure 5 presents the value of human lives lost due to suicide in high-income and upper-middle-income countries (Group 1).

### 3.3. Money Value of Human Life Lost Among Group 2 Countries

The 34,688 SDs in Group 2 countries resulted in YLL with a value of Int$ 3,799,929,726 in 2019, or 0.11% of the group’s total GDP. The money value of YLL varied from Int$ 304,456 in Sao Tome and Principe to Int$ 1,272,197,958 in Egypt. Nearly 48.74% of the value of YLL was incurred by Egypt and Ghana. Figure 6 portrays the value of YLL due to suicide in lower-middle-income economies (Group 2).

### 3.4. Money Value of Human Life Lost Among Group 3 Countries

The 31,537 SDs in Group 3 countries resulted in YLL valued at Int$ 998,610,772 in 2019, or 0.12% of the group’s total GDP. The value ranged from Int$ 3,309,196 in Central African Republic to Int$ 246,143,613 in Uganda. Ethiopia and Uganda bore 41.64% of the group’s value of YLL. Although Group 3 had 22,257 more SDs than Group 1, the value of YLL by Group 1 was greater than that of Group 3 by Int$ 1,175,220,150. Figure 7 shows the money value of YLL due to suicide in low income economies (Group 3).

### 3.5. Average Value of Human Lives

The mean value of YLL per SD and per person in the population for the 54 countries are contained in Table 3.

Table 4 displays the total present value, average present value per SD, and average present value per person in population for Groups 1–3. The monetary value per human life lost from SD was Int$ 234,244 for Group 1, Int$ 109,545 for Group 2 and Int$ 32,223 for Group 3. The mean value of YLL per person in population was Int$ 18.6 for Group 1, Int$ 6.2 for Group 2 and Int$ 1.9 for Group 3. The mean value of YLL per SD in Group 1 was about two-fold that of Group 2 and seven-fold that of Group 3.

One of the key drivers of expected value of YLL is the size of per capita GDP. For example, although number of SD in Group 1 countries such as Equatorial Guinea and Seychelles were only 66 and 9, respectively, the value of YLL per SD for the two countries were substantial at Int$ 410,550 and Int$ 2,659,224. Group 3 with comparatively higher number of SD such as for the Democratic Republic of the Congo with 5430 deaths and Ethiopia with 5106 deaths have comparatively lower values of YLL per SD of Int$ 9566 and Int$ 33,226, respectively.

### 3.6. Sensitivity Analysis Results

Impact of changes in discount rates: The use of discount rate of 5% led to a reduction in the grand total value of YLL due to SD of Int$ 1,595,216,155 (22.8%) and the mean value per SD by Int$ 21,127. Application of discount rate of 10% reduced the grand total value of YLL by Int$ 3,684,299,083 (51.7%) and the mean value of YLL per SD by Int$ 48,795.

Impact of changes in life expectancy: Re-estimation of the 19 equations with highest in the African continent (that of Algeria), instead of individual countries average life expectancies, increased the grand total value of YLL by Int$ 2,316,919,656 (33.1%) and the mean value of YLL per SD by Int$ 30,686. Utilisation of the average life expectancy of Japanese females increased the grand total value of YLL by Int$ 4,381,129,424 (62.7%) and the average value of YLL per SD by Int$ 58,025.

## 4. Discussion

The estimated total value of YLL due to SD was approximately 0.12% of the 2019 GDP of the 54 African countries. Both discount rate and life expectancy assumed were found to be the key determinants of the total value of YLL from SD.

How does the cost of SD in Africa compare with that of other regions? In North America, Shepard et al. [23] estimated the total economic burden of suicide and suicide attempts in the USA in 2013 at $ 93.5 billion, of which 97.1% was lost productivity. Based on 2010 data, CDC estimated that suicide costs the USA society over $ 44.6 billion a year [22]. Yang and Lester [25] estimated the USA economic cost of suicide and non-fatal attempts to be $ 16 billion and $ 4.7 billion, respectively, in 2005. Corso et al. [26] estimated the lifetime medical costs and productivity losses associated with interpersonal and self-directed violence in USA (in 2000) to be $ 64.8 billion, of which 99.4% was due to indirect lost.

In Europe, Orlewska and Orlewska estimated the economic burden of suicide to be € 694.6 million in Poland in 2012 [29]. Rivera et al. [30] estimated that suicide in Spain in 2013 resulted in an economic loss of over € 565 million. Kennelly [34] estimated the total cost of suicide in Ireland to be € 906 million in 2001 and € 835 million in 2002. Platt et al. [35] estimated the economic cost of suicide in Scotland at £ 1,079,601,213 in 2004.

In the Western Pacific, studies appraised the economic cost of suicide on the Australia’s economy to be approximately $ 17.5 billion per year [44]; New Zealand’s economy at $ 2.187 billion [45]; Taiwan’s economy at TWD 32.5 billion (US$ 1.084 billion); and Japan’s economy at US$ 2.542 billion [50].

In Southeast Asia, studies assessed the cost of suicide on Thailand’s economy to be Baht 12.77 billion (US$ 422.2 million) [52] and India’s at Indian Rupees 34.884 billion (US$ 491.711 million) [81].

The African continent average cost per SD was lower than the average cost per suicide of £1.67 million in England [32], £ 1.5 million in Ireland [34] and £ 1.3 million in Scotland [35].

How does the cost of SD compare with those of other causes death in Africa? The first study estimated the total indirect cost of diabetes deaths that occurred in the WHO African Region (AFR) in 2000 to be Int$ 1.69 billion and cost per diabetes death of Int$ 107,664 [63]. The second study calculated the total cost of cholera-related deaths that occurred in 2005 to be Int$ 124.4 million and the cost per death to be Int$ 54,414 [64]. The third study estimated the total cost of maternal deaths that occurred in 2010 to be Int$ 4.462 billion and the cost per death to be Int$ 30,203 [65]. The fourth study appraised the total cost of maternal deaths that happened in 2013 to be Int$ 5.53 billion and the cost per death to be Int$ 32,934 [66]. The fifth study assessed the total cost of child mortality in 2013 to be Int$ 150.3 billion and the cost per death to be Int$ 50,494 [67]. The sixth study calculated the total productivity losses due to EVD deaths in West Africa to be Int$ 155.663 million and the cost per death to be Int$ 13,856 [69]. The seventh study valued the human lives lost due to Ebola Virus Disease (EVD) in the Democratic Republic of the Congo to be Int$ 17.761 million and the cost per death to be Int$ 13,801 [69]. The eighth study evaluated the total cost of tuberculosis (TB) deaths in 2014 to be Int$ 50.4 billion and the cost per death to be Int$ 66,872 [70]. The ninth study estimated the total cost of NCD deaths in 2012 to be Int$ 61.3 billion and the cost per death to be Int$ 21,985 [71]. The tenth study assessed the total deaths from neglected tropical diseases (NTD) in 2015 to be Int$ 5.113 billion and the cost per death to be Int$ 75,339 [72]. Therefore, the continental Africa cost per SD of Int$ 92,576 was greater than that of maternal death, child death, EVD death, TB death, NCD death, NTD death and cholera death. However, cost per SD was less than that of diabetes mellitus death.

### 4.1. Potential Use of the Evidence

The estimates reported in this paper show the maximum possible economic cost reduction if suicides were prevented. The results can potentially be used for: (a) raising the public and policy-maker’s awareness of the potential economic impact of completed suicides; (b) developing an investment case for mental health interventions geared at reducing depression, which is a major cause of suicide attempts; and (c) use by Ministries of Health and other stakeholders in advocating for prioritisation of suicide prevention on the continental, sub-regional and national development and public health policy agendas.

### 4.2. Limitations

This study has a number of limitations that readers should be aware of. First, due to underreporting attributed to weaknesses of surveillance systems [18] and stigma surrounding suicide, reported numbers are likely to be higher. Therefore, the GDP losses reported in this paper are likely to be lower than the actual losses associated with premature mortality from suicide in the continent.

Second, estimation of economic consequences of non-fatal suicide attempts was beyond the scope of the analysis reported in this paper. Our study omits the health service costs (inpatient stay, intensive unit care and psychological assessment) related to suicide attempts [82]. However, suicide attempts often lead to serious physical injuries (broken bones, brain damage or organ failure) and mental health problems such as depression which undermine performance of activities of daily living [6,83]. Marcotte [84] postulated most suicide attempts are not fatal; thus, estimation of only the cost of completed (fatal) suicide might lead to significant underestimation of the total cost of suicide and suicide attempts in the continent.

Third, the cost of time lost by family, workmates, friends and community in meetings to organise for burial and attending funeral ceremonies was not factored into the current study. In African context, funerals are usually big events, which are attended by hundreds of people.

Fourth, other direct costs (autopsy, transportation, burial, food and beverages eaten by those attending funeral ceremonies, legal, etc.) associated with suicides were omitted.

Fifth, the study does not capture the psychological trauma (emotional suffering) associated with grief reactions (guilt, anger, abandonment, denial, helplessness and shock) and depression among family members (i.e., children, siblings, parents and grandparents), relatives, friends, colleagues and wider community [4,85].

Sixth, we did not estimate the potential savings to the society from: “(a) not having to treat depression and other psychiatric disorders of those who kill themselves; (b) avoidance of pension, social security and nursing home care costs; and (c) assisted suicide (where legalised)” (p.351) [25]. Therefore, our estimates are not net of such potential savings.

## 5. Conclusions

In this study, we made a germinal attempt to estimate the discounted stream of monetary value of potential YLL due to suicide in all African countries. It is clear that suicide imposes a substantive economic burden on African economies. Sensitivity analysis revealed that the magnitude of money value of YLL largely depends on the discount rate and average life expectancy assumed. This evidence of forgone monetary value of human life, in addition to human rights arguments, reinforces the case for increased multi-sectoral investments to ensure universal coverage of promotive, preventive, curative and rehabilitative mental health services in the continent.

There is urgent need for all African Union Heads of State and Government to ceaselessly beat rhythmically the African drums of war against mental disorders, including the scourge of depression, which fuel suicide. The goal would be to energise all domestic and external stakeholders to wage spirited synchronised strategic manoeuvres against the root causes of mental disorders in the African cultural, geopolitical and socioeconomic contexts. That war will have to be underpinned by various forms of evidence to ensure that the health impact of all available public and private resources is maximised.

Since the analysis reported in this paper was a partial cost-of-illness study, it calls for studies of total economic burden of fatal and non-fatal suicide attempts to provide a full picture of suicide-related economic consequences in the continent for use in advocacy. In addition, there is need for economic studies that would yield pertinent evidence to guide national policy and planning. Such studies include use of extended utility maximisation theories to explain why some individuals in African populations engage in suicide attempts [84,86,87]; economic evaluation (cost–benefit analysis, cost–utility analysis and cost-effectiveness analysis) of multi-sectoral social policy interventions for preventing and managing suicide attempts; and national mental health accounts to monitor resources currently being invested by national governments and other stakeholders in tackling root-causes of suicide attempts in Africa.

## Figures and Tables

**Figure 1 healthcare-08-00084-f001:**
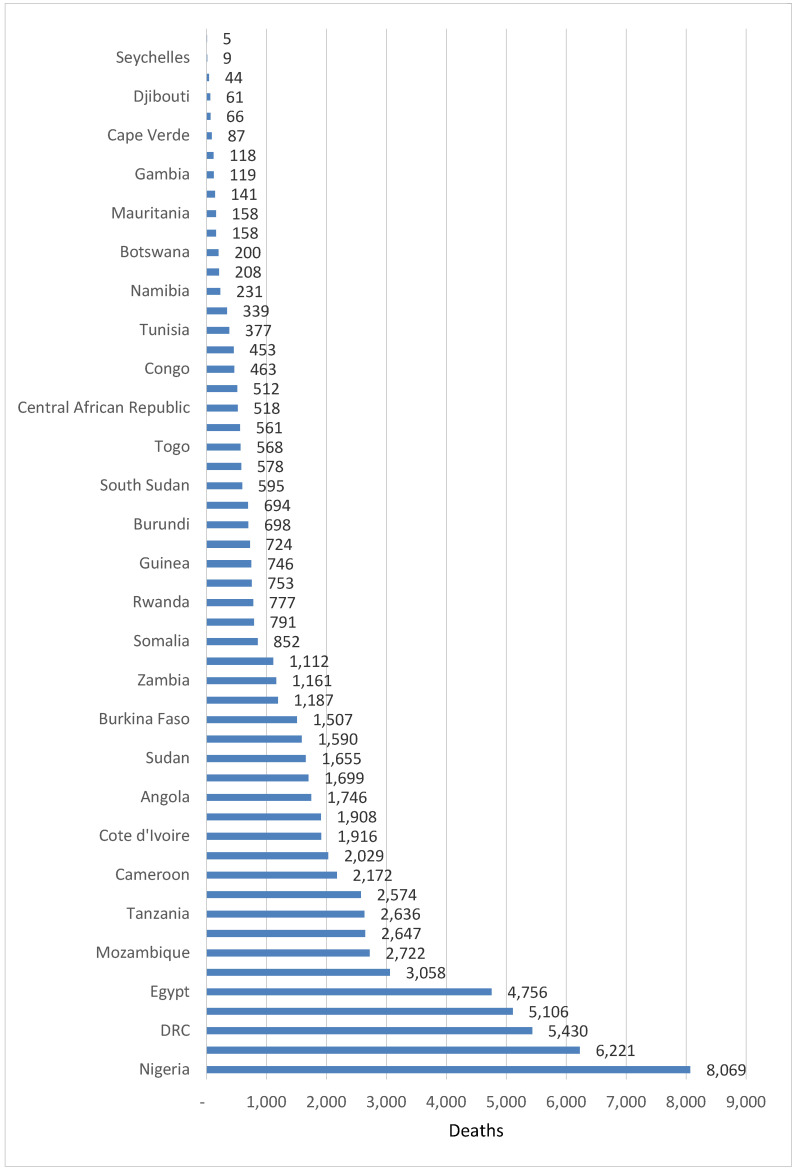
Number of suicide deaths in countries of the African continent in 2017.

**Figure 2 healthcare-08-00084-f002:**
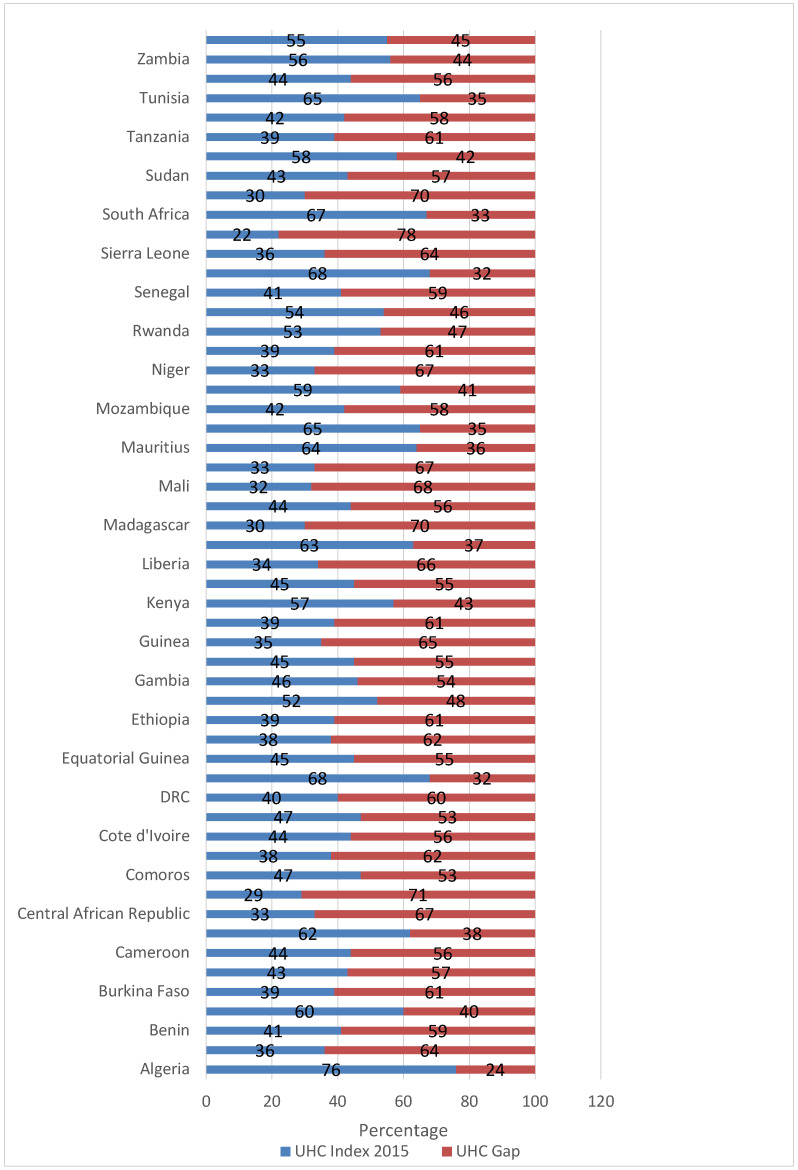
Universal health coverage indices and gaps among African countries, 2015.

**Figure 3 healthcare-08-00084-f003:**
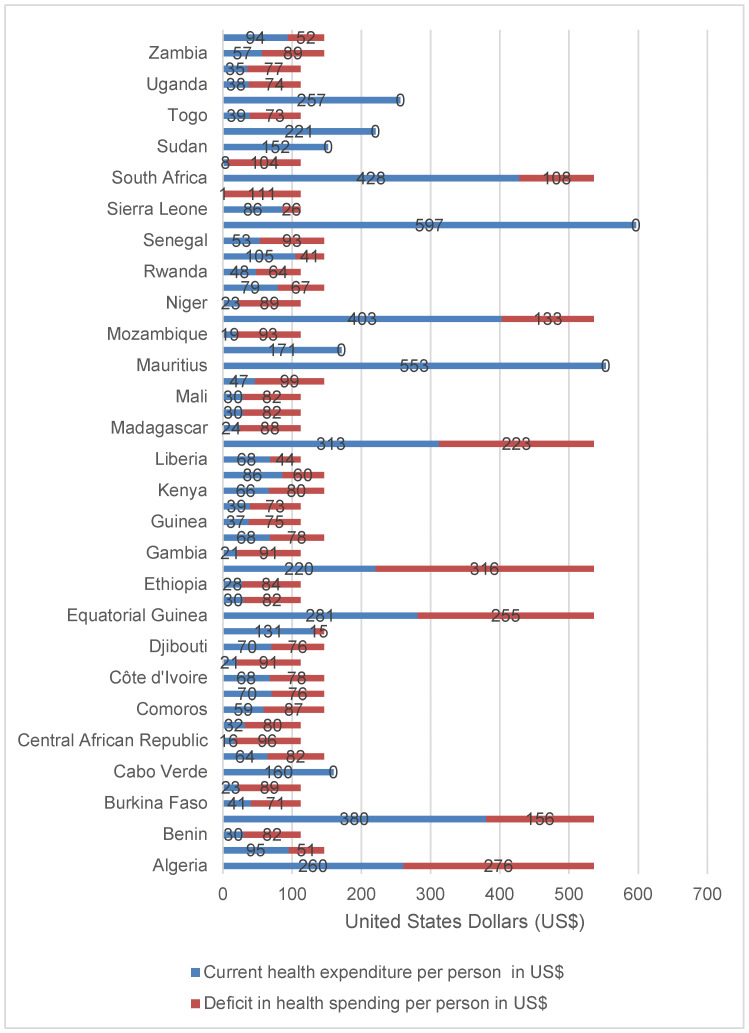
Current health expenditure per capita and deficit in health spending in Africa, 2016.

**Figure 4 healthcare-08-00084-f004:**
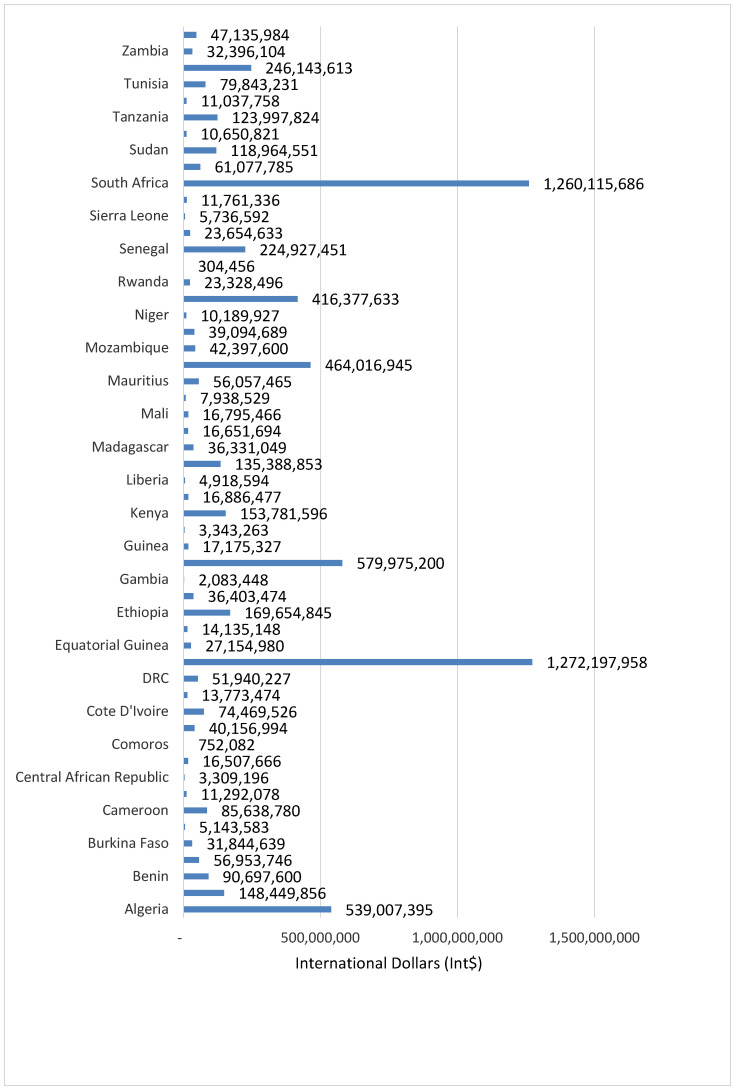
Monetary value of human lives lost due to suicide in 54 countries in Africa (Int$, in 2019).

**Figure 5 healthcare-08-00084-f005:**
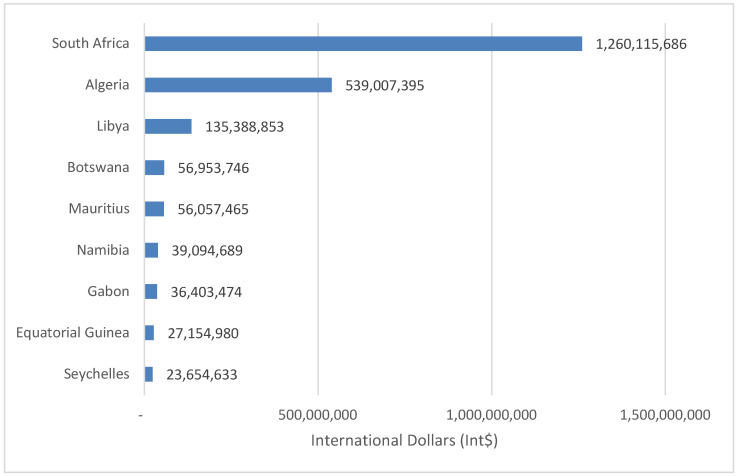
Monetary value of human lives lost due to suicide in high-income and upper-middle-income countries (Group 1) of Africa (Int$, in 2019).

**Figure 6 healthcare-08-00084-f006:**
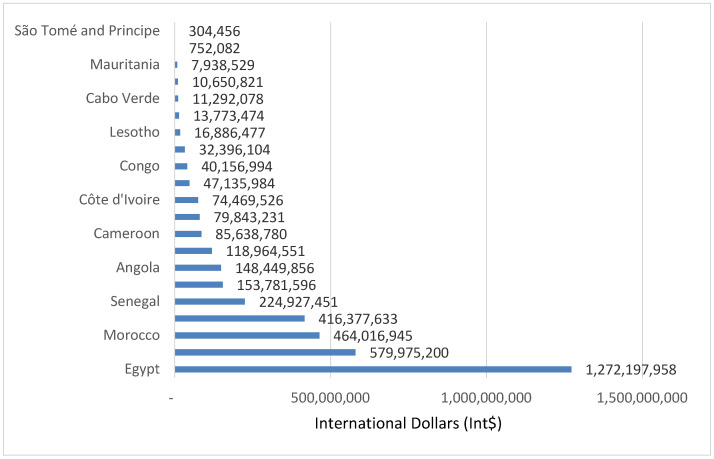
Monetary value of human lives lost due to suicide in lower-middle-income countries (Group 2) of Africa (Int$, in 2019).

**Figure 7 healthcare-08-00084-f007:**
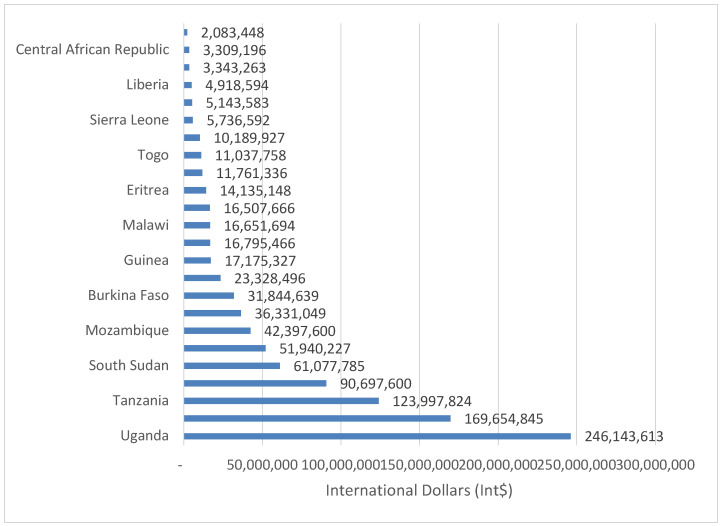
Monetary value of human lives lost due to suicide in low-income countries (Group 3) of Africa (Int$, in 2019).

**Table 1 healthcare-08-00084-t001:** African continent countries by income classification.

Income Group	Countries
Group 1: High income (US$ 12,375 or more) and Upper middle income (US$ 3996–12,375)	Seychelles, Algeria, Botswana, Equatorial Guinea, Gabon, Libya, Mauritius, Namibia, South Africa (*n* = 9)
Group 2: Lower middle income (U$ 1026–3995)	Angola, Cabo Verde, Cameroon, Comoros, Congo, Côte d’Ivoire, Djibouti, Egypt, Eswatini (Swaziland), Ghana, Kenya, Lesotho, Mauritania, Morocco, Nigeria, São Tomé and Principe, Senegal, Sudan, Tunisia, Zambia, Zimbabwe (*n* = 21)
Group 3: Low income (US$ 1025 or less)	Benin, Burkina Faso, Burundi, Central African Republic, Chad, Democratic Republic of Congo, Eritrea, Ethiopia, The Gambia, Guinea, Guinea-Bissau, Liberia, Madagascar, Malawi, Mali, Mozambique, Niger, Rwanda, Sierra Leone, Somalia, South Sudan, Tanzania, Togo, Uganda (*n* = 24)

Source: World Bank [1].

**Table 2 healthcare-08-00084-t002:** Potential pecuniary value of human life lost per age group (2019, Int$ or PPP).

Age Group	Monetary Value (Int$)	Percent (%)
10–14	165,278,172	2.36
15–19	629,799,693	9.01
20–24	1,014,518,001	14.51
25–29	1,023,659,026	14.64
30–34	1,857,591,488	26.58
35–39	784,496,461	11.22
40–44	559,357,213	8.00
45–49	411,995,125	5.89
50–54	287,655,819	4.12
55–59	161,681,640	2.31
60–64	69,517,406	0.99
65–69	20,064,418	0.29
70–74	4,348,864	0.06
Total	6,989,963,325	100.00

**Table 3 healthcare-08-00084-t003:** Discounted monetary value of human lives due to suicide in countries of continental Africa (Int$ or PPP, in 2019).

Country	(A) Population	(B) SD	(C)Total Value of Human Lives Lost Due to SD (Int$)	(D) Value of Human Life Lost per SD (Int$) [D = (C/B)]	(E) Value of Human Life Lost Per Person in Population (Int$) [E = (C/A)]
Algeria	43,088,000	1699	539,007,395	317,312	12.51
Angola	30,053,000	1746	148,449,856	85,023	4.94
Benin	11,722,000	694	90,697,600	130,742	7.74
Botswana	2,378,000	200	56,953,746	284,592	23.95
Burkina Faso	19,996,000	1507	31,844,639	21,126	1.59
Burundi	11,529,000	698	5,143,583	7374	0.45
Cameroon	25,506,000	2172	85,638,780	39,433	3.36
Cape Verde	551,000	87	11,292,078	129,637	20.49
Central African Republic	5,181,000	518	3,309,196	6389	0.64
Chad	12,802,000	791	16,507,666	20,863	1.29
Comoros	872,000	44	752,082	17,266	0.86
Republic of Congo	4,568,000	463	40,156,994	86,805	8.79
Cote d’Ivoire	26,275,000	1916	74,469,526	38,870	2.83
Democratic Republic of Congo	91,931,000	5430	51,940,227	9566	0.56
Equatorial Guinea	887,000	66	27,154,980	410,550	30.61
Eritrea	6,159,000	561	14,135,148	25,213	2.30
Ethiopia	95,644,000	5106	169,654,845	33,226	1.77
Gabon	2,080,000	158	36,403,474	230,189	17.50
Gambia, The	2,238,000	119	2,083,448	17,473	0.93
Ghana	29,742,000	1908	579,975,200	304,007	19.50
Guinea	13,627,000	746	17,175,327	23,015	1.26
Guinea-Bissau	1,776,000	141	3,343,263	23,777	1.88
Kenya	49,364,000	3058	153,781,596	50,282	3.12
Lesotho	2,048,000	512	16,886,477	33,003	8.25
Liberia	4,978,000	339	4,918,594	14,492	0.99
Madagascar	27,055,000	1590	36,331,049	22,846	1.34
Malawi	20,289,000	1187	16,651,694	14,023	0.82
Mali	20,161,000	724	16,795,466	23,192	0.83
Mauritania	4,058,000	158	7,938,529	50,349	1.96
Mauritius	1,279,000	118	56,057,465	476,497	43.83
Mozambique	31,157,000	2722	42,397,600	15,577	1.36
Namibia	2,408,000	231	39,094,689	169,130	16.24
Niger	19,939,000	753	10,189,927	13,539	0.51
Nigeria	199,206,000	8069	416,377,633	51,599	2.09
Rwanda	12,432,000	777	23,328,496	30,036	1.88
Sao Tome and Principe	222,000	5	304,456	57,046	1.37
Senegal	16,793,000	1112	224,927,451	202,362	13.39
Seychelles	96,000	9	23,654,633	2,659,224	246.40
Sierra Leone	7,737,000	453	5,736,592	12,654	0.74
South Africa	58,333,000	6221	1,260,115,686	202,558	21.60
South Sudan	13,378,000	595	61,077,785	102,569	4.57
Swaziland	1,177,000	208	10,650,821	51,284	9.05
Tanzania	52,067,000	2636	123,997,824	47,044	2.38
Togo	8,205,000	568	11,037,758	19,431	1.35
Uganda	40,007,000	2029	246,143,613	121,334	6.15
Zambia	18,321,000	1161	32,396,104	27,908	1.77
Zimbabwe	15,658,000	2647	47,135,984	17,804	3.01
Djibouti	1,078,000	61	13,773,474	226,015	12.78
Egypt	99,211,000	4756	1,272,197,958	267,508	12.82
Libya	6,578,000	578	135,388,853	234,068	20.58
Morocco	35,587,000	2574	464,016,945	180,238	13.04
Somalia	15,540,481	852	11,761,336	13,797	0.76
Sudan	43,222,000	1655	118,964,551	71,863	2.75
Tunisia	11,783,000	377	79,843,231	211,807	6.78

**Table 4 healthcare-08-00084-t004:** Present value of YLL due to suicide in Africa (Int$ or PPP, in 2019).

Summary of Indirect Costs	Group 1: High-Income and Upper-Middle-Income Countries Sub-Total Cost (Int$)	Group 2: Lower-Middle-Income Countries Sub-Total Cost (Int$)	Group 3: Low-Income Countries Sub-Total Cost (Int$)	Grand total Cost (Int$)
(1) Total present value of SDs	2,173,830,922	3,799,929,726	1,016,202,676	6,989,963,325
(2) Average present value per SD	234,244	109,545	32,223	92,576
(3) Average present value per person in population	18.56	6.18	1.86	5.5
% of grand total	31.0	54.5	14.5	100

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
