# Peer review of "The Monetary Value of Human Lives Lost to Suicide in the African Continent: Beating the African War Drums"

_healthcare, 2020, doi:10.3390/healthcare8020084_

Round 1
Reviewer 1 Report
The manuscript is a study that addresses the cost of suicide in African countries.
The introduction of the manuscript is adequate for the aim.
The method is well described, but some missing full name of abbreviations.
The results are clearly exposed, but the graphics are not completed. Some countries’ names are missed.
The discussion, the conclusions and the limitations of the study are adequate to the results and the aim.
Reviewer 1 Report
Reviewer 1 comment: The introduction of the manuscript is adequate for the aim
Authors response: Thanks. No response required.
Reviewer 1 comment: The method is well described, but some missing full name of abbreviations.
Authors response: All the abbreviations are now written in full where they first appear.
Reviewer 1 comment: The results are clearly exposed, but the graphics are not completed. Some countries’ names are missed.
Authors response: Wherever the names of some countries are missing in graphs, it's where data was not available.
Reviewer 1 comment: The discussion, the conclusions and the limitations of the study are adequate to the results and the aim.
Authors response: No response required.
Reviewer 2 Report
I think this paper should be accepted after minor revisions. The graphs need to ensure they are publication worthy and fully labelled. Figure 5 could be deleted or turned into a bar graph. The under reporting of suicide is acknowledged by the authors. I think they need to do a better job in the Discussion of describing why this work is important. What can we do with this information now that we have it? The authors could reduce the number of graphics in the paper without losing the main points. In fact, many of the graphs could be added to supplemental information to accompany the paper, but not be part of the main paper. The authors should review them and decide which are truly core to the paper's objectives. The format made it difficult to read as the graphs at the beginning interrupted the flow and are all good candidates for movement to supplemental information.Author Response
Reviewer 2 report
Comments and Suggestions for Authors
I think this paper should be accepted after minor revisions.
Reviewer 2 comment: The graphs need to ensure they are publication worthy and fully labelled.
Authors response: All the figures are relevant to the objectives of the manuscript. They are all fully labelled now; and the titles have been placed at the bottom of each figure as recommended by the journal.
Reviewer 2 comment: Figure 5 could be deleted or turned into a bar graph.
Authors response: Figure 5 has been converted from pie chart to a bar graph as suggested. See Lines 278-279.
The Figure 5 presents the monetary value of human lives lost due to suicide in high-income and upper-middle-income countries (Group 1) of Africa. It is crucial to retain the Figure since the analysis is by economic classification of countries.
Reviewer 2 comment: The authors could reduce the number of graphics in the paper without losing the main points. In fact, many of the graphs could be added to supplemental information to accompany the paper, but not be part of the main paper. The authors should review them and decide which are truly core to the paper's objectives.
Authors response: (1). Figure 1 entitled “Figure 1. Number of suicide deaths in countries of the African continent in 2017” is necessary since the focus of the paper is the number of suicide deaths per country in Africa. It gives the reader a pictorial idea of the magnitude of the problem.
Figure 2 entitled “Figure 2. Universal health coverage (UHC) indices and gaps among African countries, 2015” shows the gaps in essential health services, including mental health services that are meant to address the factors that lead people to take their lives. Thus, it helps to justify the need for advocacy for increased investments into scaling up coverage of essential health services.
Figure 3 entitled “Current health expenditure per capita and deficit in health spending in Africa, 2016” shows the shortfalls in health expenditure per person. Again building the case for need to increase investments in health systems to tackle health challenges, including depression (which is a major cause of suicide).
Figure 4 to 7 present the results of our analysis by each of the three economic groupings.
Therefore, we would request the editor to retain all the seven Figures.
Reviewer 2 comment: The format made it difficult to read as the graphs at the beginning interrupted the flow and are all good candidates for movement to supplemental information.
Authors response: The figures have been moved to the end of the relevant paragraph to facilitate flow.
-Figure 1 – see Lines 47-52
-Figure 2 – see Lines 73-78
-Figure 3 – see Lines 87 – 91
-Figure 4 – No change. It is in appropriate place.
-Figure 5 – see Lines 275 - 279
-Figure 6 – see Lines 285 - 289
-Figure 7 – see Lines 295 - 299
Reviewer 2 comment: I think they need to do a better job in the Discussion of describing why this work is important. What can we do with this information now that we have it?
Authors response: In Discussion section, we have included a subsection entitled “Potential use of the evidence”. The subsection can be found in Lines 372-380.